# Sex-specific associations between person and environment-related childhood adverse events and levels of cortisol and DHEA in adolescence

Lotte van Dammen[1,2,3]*, Susanne R. de Rooij[4,5], Pia M. Behnsen[5], Anja C. Huizink[5,6]

**1** Department of Human Development & Family Studies, Iowa State University, Ames, Iowa, United States of America, **2** Department of Obstetrics and Gynaecology, University of Groningen, University Medical Center Groningen, Groningen, the Netherlands, **3** Department of Epidemiology, University of Groningen, University Medical Center Groningen, Groningen, the Netherlands, **4** Department of Clinical Epidemiology, Biostatistics and Bioinformatics, Amsterdam UMC, location AMC, Amsterdam, The Netherlands, **5** Department of Developmental Psychology, VU University Amsterdam, Amsterdam, The Netherlands, **6** School of Health and Learning, University of Skövde, Skövde, Sweden

* lotte@iastate.edu

**Data Availability Statement:** All relevant data are within the manuscript and its supporting information files.

## Abstract

### Background

Person and environment-related childhood adverse events have been demonstrated to increase the risk of impaired mental health in later life differently for boys and girls. Altered hypothalamic pituitary adrenal (HPA)-axis functioning has been suggested as a key mechanism underlying this association. Cortisol and dehydroepiandrosterone (DHEA) are both output hormones of the HPA-axis. DHEA may have a protective function against long-term exposure to increased levels of cortisol, but has been little investigated in relation to childhood adversity.

### Objective

We aimed to test the associations between person-, and environment-related childhood adversity and levels of cortisol, DHEA and cortisol/DHEA ratio in adolescent boys and girls.

### Methods

A total of 215 Dutch adolescents participated in the study and filled out the 27-item Adverse Life Events Questionnaire for the assessment of childhood adversity, which was split up in separate scores for person-related and environment-related events. Cortisol and DHEA concentrations and cortisol/DHEA ratio were determined in proximal 3 cm long hair segments. Additionally, saliva samples were collected immediately and 30 minutes after waking up, at noon and at 8 pm. Multiple linear regression analyses were used to test associations between childhood adversity and cortisol and DHEA concentrations, for boys and girls separately, with age, BMI and pubertal development as covariates.

**Funding:** The authors received no specific funding for this work.

**Competing interests:** The authors have declared that no competing interests exist.

## Results

Data were available for 74 boys and 116 girls with a mean age of 15.7 years (SD = 2.0). Higher levels of person-related childhood adversity were associated with higher hair DHEA levels in girls and with higher hair cortisol levels in boys. A trend towards a significant association was observed between higher levels of environment-related childhood adversity and higher DHEA levels in boys. Neither person- nor environment related childhood adversity was associated with cortisol/DHEA ratio. A trend was observed for environment-related childhood adversity and lower daily cortisol output in boys.

## Conclusion

We found differential associations between childhood adversity and cortisol and DHEA levels in girls and boys, for respectively person-related and environment-related childhood adversity. Our findings suggest that different types of childhood adversity are not only linked to levels of cortisol, but also to DHEA concentrations, in a sex-specific manner, with possible future implications for mental health.

## Introduction

Childhood adversity refers to a range of negative exposures during childhood, including abuse, neglect, violence, parental separation or incarceration, and occurs on a common basis [1]. In high income countries, the prevalence of having experienced at least one adverse event during childhood has been estimated to be almost 40% [2]. Childhood adversity can have detrimental effects on later mental health, leading to depression, substance abuse and suicide attempts in adulthood [1, 3, 4].

Childhood adversity can be divided into dependent or person-related events and independent or environment-related events [5, 6]. Person-related events have a direct impact on the person himself or herself. Examples of such events are severe physical illness, a handicap, being the victim of physical or sexual abuse. Environment-related events affect other individuals or the direct environment of an individual. Such events include parental separation, parental drug use or the death of a sibling or parent. It has been previously shown that these distinct categories of events can affect mental health in a distinct manner [6]. Person-related childhood adverse events seem to mainly relate to depressive symptoms. In a co-twin control study, person-related childhood adverse events were associated with the onset of depression among females more than environment related events [7]. Another two studies showed more depressive symptoms among adolescents who experienced person-related adverse events, but not among those with environment-related events [8, 9]. Environment-related childhood events, including parental separation and sudden parental death, have been associated with more general psychological maladjustment and conduct problems [10, 11].

The associations between childhood adversity and mental health outcomes might be partly explained by effects of childhood adverse events on the development of the hypothalamic pituitary adrenal (HPA)-axis in boys and girls. The main end product of the HPA-axis is cortisol, which is released after stress exposure and is involved in processes such as glucose release and immune responsivity. In addition to release after stress, cortisol follows a diurnal rhythm with peak levels shortly after awakening and decreasing levels during the day [12]. Dysregulation of HPA-axis functioning has been associated with depressive symptoms, anxiety disorders and PTSD in childhood and adulthood [13–15]. Another important HPA-axis related hormone is

dehydroepiandrosterone (DHEA), a human steroid and androgen that is produced in the adrenal cortex. Levels of DHEA typically increase when cortisol levels increase. DHEA has antiglucocorticoid properties and the co-release of DHEA with cortisol returns the stress system back to homeostasis, suggesting a protective function against long-term exposure to increased levels of cortisol [16, 17]. Low, but also high DHEA concentrations have been associated with an increased prevalence of depression and PTSD in adults [18–21]. An increased ratio of cortisol to DHEA concentration suggests higher stress hormone levels and lower 'protective' hormone levels, and has been related to poor mental health outcomes [18].

Associations between adversity during childhood and altered HPA-axis functioning have frequently been observed in animal and human studies [22, 23]. Adverse childhood events have been linked to blunted diurnal cortisol patterns in young children [24] and adolescents [25] and an increased cortisol awakening response in adult women [26]. Childhood adverse events such as childhood maltreatment have also been linked to increased cortisol/DHEA ratio concentrations in adulthood [27] emphasizing the complexity of associations between childhood adversity and neuroendocrine variables.

The association between childhood adversity and DHEA or cortisol/DHEA ratio has not been studied previously in adolescent boys and girls despite our understanding of the adolescent period as an important developmental period, including major changes in the HPA-axis and associated adaptability to stress. These changes have been linked to increased risk for the onset of internalizing and externalizing behavior problems among children and adolescents [24, 28]. DHEA plays a role in the onset of adrenarche, the physical maturation as well as brain development during puberty [28]. Also, to the best of our knowledge, associations between person- and environment-related childhood adverse events specifically and HPA-axis measures have not yet been studied. In order to further unravel the associations between different types of childhood adversity and cortisol, DHEA, cortisol/DHEA ratio in adolescence, the present study was carried out. We expected associations between both environment-related and person-related childhood adversity and cortisol and DHEA levels, but did not hypothesize on the direction of associations due to the inconsistency in the current literature on direction of associations between adversity and HPA-axis outcomes. As there are indications for differences between boys and girls in HPA-axis functioning during adolescence, we tested the associations separately for boys and girls.

## Methods

### Participants

Participants (age 12–18) were recruited from several high schools in the Netherlands. A total of 215 adolescents consented to participation in the study. The availability of questionnaire data, hair (part 1) and saliva (part 2) data is shown in Fig 1.

The study was approved by the ethical committee of the Faculty of Behavioral and Movement Sciences of the Vrije Universiteit Amsterdam (study number: VCWE-2014-100). Participants, and parents when children were under the age of 16 years, gave written informed consent.

### Procedure

Study assistants asked permission of high schools to invite students to participate in the study. When permission was obtained, youths and their parents received a written explanation of the study with an invitation to participate. The student assistants then visited schools to enroll adolescents willing to participate and who had filled out a written informed consent (by themselves or from their parents). After explaining the study protocol, a piece of hair was cut as the first part of the study. In the second part of the study, within a week after the first part of the

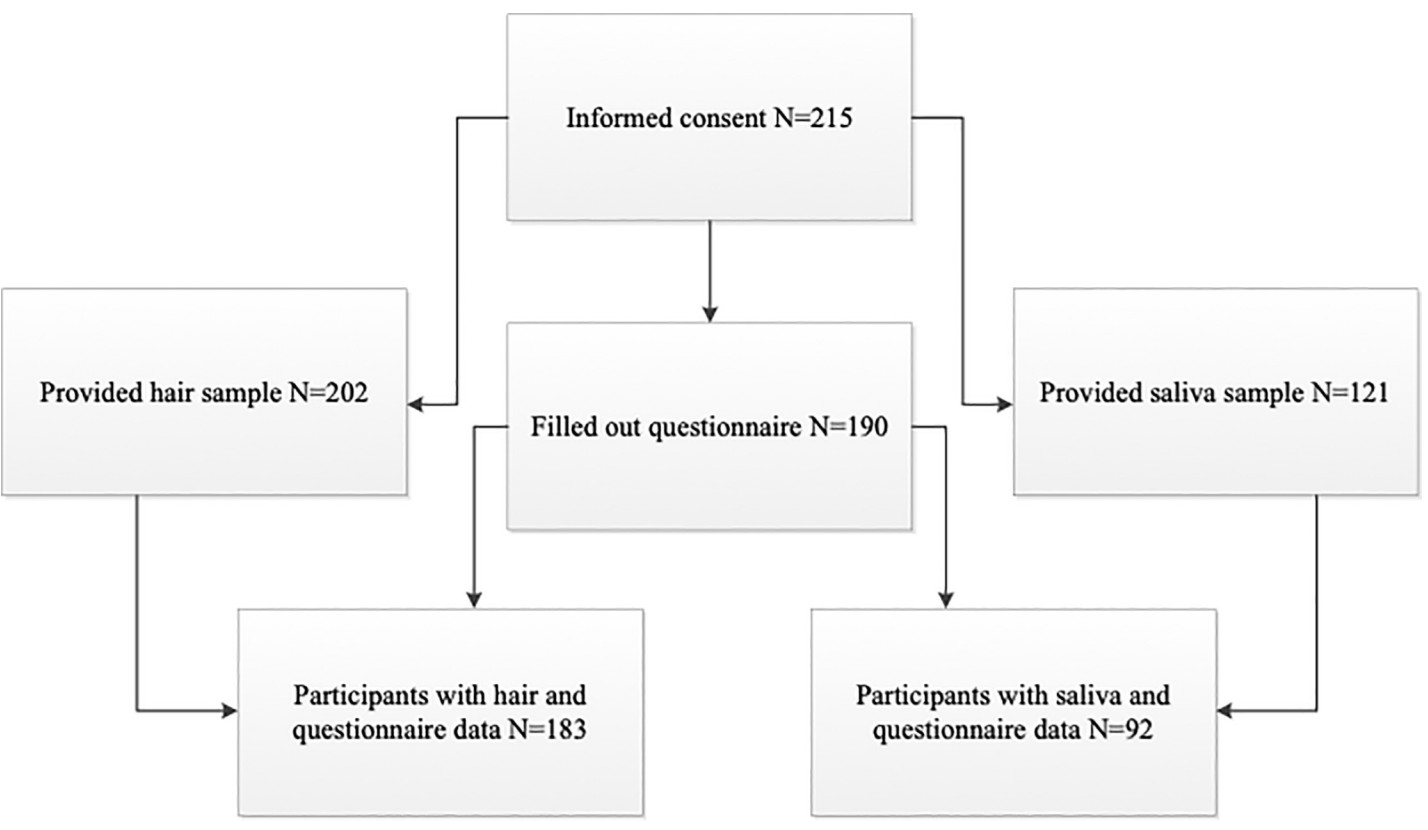

**Fig 1. Flowchart of participants and data availability for questionnaires, hair (part 1) and saliva (part 2).**

study, half of the subjects received four tubes to collect saliva on a weekly day along with information on saliva collection. They were asked to collect the saliva in the tubes and return the tubes to the university by mail. The time of day when each saliva sample had to be collected was explained in the information letter, and participants reported the exact time of collection of each saliva sample. After this, participants were emailed a link to a website where they were asked to fill out a questionnaire containing questions on demographic variables and seven different standardized questionnaires (see below for a description of the questionnaires used for the present study). Completing the questionnaires took about one hour.

## Self-report measures

The questionnaire contained questions about height and weight, ethnic background, and asked participants whether they had ever smoked. Postal code was also self-reported, upon which socio-economic status was based. The postal code was recoded to a status score from 2016 that was developed by the Netherlands Institute for Social Research [29]. The status score reflects the social status of a neighborhood compared to other neighborhoods, with positive scores representing higher socio-economic status, relative to the Dutch average of 0 (range in the current sample: -3.45 to 2.61).

The Pubertal Development Scale (PDS) was used to assess the stage of puberty for boys and girls. The PDS is a 5-item questionnaire regarding growth, bodily hair development, skin changes and sex-specific pubertal development questions about voice changes and facial hair (boys) and breast development and menarche (girls). A total score between 1 (no development) and 4 (development complete) was computed ($\alpha = 0.77$) [30, 31].

Childhood adversity can be studied by using a questionnaire assessing stressful life events, containing items about different types of events. In several studies these events have then been summed to a total number of events per person [27, 32]. However, the death of a parent may have a much greater impact on the life of a child or adolescent, compared to the loss of a job by a parent. In order to take the potential impact of an event into account, questionnaires like the Holmes and Rahe Stress Scale use life change units have been developed [33]. For the present study, we used the 27-item Adverse Life Events Questionnaire (ALEQ), a reliable and valid measure of childhood adversity, and also took the impact of events into account. For each adverse event, the participant filled out if he/she experienced the event, how many times, the age at the time of the adverse event, and the perceived severity [6, 34]. If the event was not perceived as severe at all, the event was not scored as a stressful event. Events were categorized and scored with the Holmes and Rahe stress scale, resulting in a total life change unit score, with higher scores indicating higher impact of stressful events [33]. We split the total score in a person-related sub score and an environment-related sub score, based on a previous study in which these events were rated by trained psychologists in the field of life events [6]. An overview of the reported person-related and environment-related is shown in Table 3.

## Neuroendocrine measures

Hair strands (about 3 mm diameter) were taken scalp-near from a posterior vertex region. Cortisol and DHEA concentrations were determined in the proximal 3 cm long hair segment which, based on an approximate hair growth rate of about 1 cm per month, is assumed to reflect hormone secretion over the three-month-period prior to hair sampling [35]. The concentrations of cortisol and DHEA were determined by liquid chromatography tandem mass spectrometry (LC-MS/MS) in the 3 one-centimeter hair samples based on previously published methods [36]. The mean concentration of cortisol and DHEA in the 3 one-centimeter hair samples represents concentrations over the last three months, whereas the mean concentration of cortisol and DHEA in the proximal one-centimeter hair sample represents concentrations over the last month. Due to the low number of 3-cm long samples, we used the concentrations of cortisol and DHEA over the last month only.

Participants were asked to passively drool saliva in the tubes at four different time points during a week day: immediately after waking up, 30 minutes after waking up, at noon (98% within one hour) and at 8 pm (94% within one hour). They were also asked to stay in bed before collecting the second sample, to not eat within the half hour preceding the saliva collection and to store the tubes with saliva in the freezer. Participants received detailed instructions, in both text and pictures, about the collection of the saliva samples. They were asked to provide the actual time of collecting the sample, also if the time deviated from the time they were supposed to collect the sample. Samples were sent unfrozen by mail and were immediately stored in the freezer upon arrival until the assays were performed. Analysis of cortisol concentrations was carried out by means of salivary cortisol immunoassay with time-resolved fluorescence endpoint detection [37]. The day area under the curve (AUCg) with respect to ground cortisol levels was computed from the salivary cortisol values as a measure of daily cortisol output. DHEA concentrations were not measured in the saliva samples. Both hair and saliva samples were processed by Dresden lab services.

## Statistical analysis

Normality was assessed by visual inspection of histograms and Kolmogorov-Smirnov tests. Non-parametric tests were used for comparison of median values. Multiple linear regression analyses were used to test the hypotheses. Because the values for cortisol, DHEA and cortisol/

DHEA ratio were not normally distributed, values were log transformed which resulted in normally distributed values. Outcomes included the hair variables DHEA over the last month, cortisol over the last month and cortisol/DHEA over the last month. Childhood adversity measured by the ALEQ in life change units was used as independent variable. Person-related and environment-related sub scores were analyzed separately. Age, BMI and PDS score were entered as covariates in the models, and each outcome was analyzed separately for boys and girls. AUCg was analyzed as a dependent variable in a regression model with dichotomized person- and environment related childhood adversity as factors. All analyses were conducted with IBM SPSS version 24 (Armonk, NY, USA).

## Results

### Descriptive analyses

The responder analyses results are shown in Table 1. The characteristics of the group that participated in the hair collection and questionnaire part of the study were similar to the characteristics of the group that participated in the saliva collection and questionnaire part.

As shown in Table 2, the final sample consisted of 190 adolescents (n = 74; 39% boys) with a mean age of 15.7 (SD 2.0) years. DHEA levels in hair, salivary cortisol levels and AUCg cortisol levels were comparable for girls and boys (all $p \geq 0.05$). Cortisol levels in hair were higher in girls (median = 3.6, inter quartile range (IQR) = 2.4–5.0) compared to boys (median = 2.4, IQR = 1.6–3.8, $p < 0.01$). Hair cortisol and DHEA were both positively associated with pubertal stage (r = 0.24, $p < 0.01$ and r = 0.22, $p < 0.01$, respectively). Hair cortisol and DHEA levels were not significantly correlated (r = 0.13, $p = 0.13$). Salivary cortisol samples and AUCg cortisol levels were not associated with pubertal stage (r range = -0.06–0.16, all $p \geq 0.05$). Comparison of hair cortisol samples and the separate saliva cortisol samples, showed that hair cortisol over the last month was significantly correlated with the midday saliva cortisol sample (r = 0.21, $p < 0.05$), but not with the first (r = -0.01, $p = 0.92$), second (r = -0.06, $p = 0.59$) and last sample (r = 0.01, $p = 0.96$). Hair cortisol over the last month was not associated with AUCg cortisol levels (r = 0.11, $p = 0.28$). Other characteristics in Table 2 were not significantly different between boys and girls (all $p \geq 0.05$).

In Table 3, the mean number and prevalence of reported life events is shown, divided into person-related events and environment-related events. For both girls and boys, the death of someone they cared about was the most common adverse event (37% and 35%, respectively), followed by severe illness or accident of a family member (21% and 24%, respectively) and having had a severe illness or accident themselves (12% and 19% respectively).

**Table 1. Responder characteristics, presented as means (SD) unless stated otherwise.**

|  | Hair & questionnaire data (n = 183) | Saliva & questionnaire data (n = 92) |
|---|---|---|
| **Age in years** | 15.6 (2.0) | 15.3 (2.1) |
| **European ethnicity, n (%)** | 163 (89.1) | 87 (94.6) |
| **Height in cm** | 170.1 (10.7) | 169.4 (10.9) |
| **Weight in kg** | 59.1 (13.1) | 58.0 (14.4) |
| **BMI in kg/m²** | 20.3 (3.2) | 20.0 (3.2) |
| **High SES n (%)** | 20 (10.9) | 11 (12.0) |
| **Smokers (ever) n (%)** | 41 (22.4) | 19 (20.7) |
| **Puberty stage** | 2.9 (0.7) | 2.8 (0.7) |

**BMI** = body mass index; **SES** = socioeconomic status.

**Table 2. Clinical characteristics, presented as means (SD) unless stated otherwise.**

| | Total sample (n = 190) | Girls (n = 116) | Boys (n = 74) |
|---|---|---|---|
| **Age in years** | 15.7 (2.0) | 15.8 (1.9) | 15.4 (2.1) |
| **European ethnicity, n (%)** | 169 (88.9) | 104 (89.7) | 65 (87.8) |
| **Height in cm** | 170.3 (10.6) | 167.4 (7.4) | 174.8 (13.0) |
| **Weight in kg** | 59.3 (13.0) | 57.8 (9.7) | 61.7 (16.6) |
| **BMI in kg/m$^2$** | 20.3 (3.2) | 20.6 (3.0) | 20.0 (3.5) |
| **High SES n (%)** | 22 (10.2) | 15 (12.9) | 7 (9.5) |
| **Smokers (ever) n (%)** | 44 (23.2) | 27 (23.3) | 17 (23.0) |
| **Life change units groups n (%)** | | | |
| None | 70 (36.8) | 41 (35.3) | 29 (39.2) |
| 1–149 | 74 (38.9) | 45 (38.8) | 29 (39.2) |
| 150–299 | 39 (20.5) | 25 (21.6) | 14 (18.9) |
| > 300 | 7 (3.7) | 5 (4.3) | 2 (2.7) |
| **Puberty stage** | 2.9 (0.7) | 3.1 (0.6) | 2.6 (0.7) |
| **DHEA pg/mg hair** * | 14.5 (9.0–25.0) | 13.9 (8.5–22.9) | 15.4 (9.4–29.5) |
| **Cortisol pg/mg hair**＊ | 3.2 (2.1–4.3) | 3.6 (2.4–5.0)** | 2.4 (1.6–3.8)** |
| **Cortisol/DHEA hair**＊ | 0.21 (0.11–0.36) | 0.26 (0.17–0.47) | 0.15 (0.08–0.28) |
| **Salivary cortisol nmol/L sample 1**＊ | 6.51 (4.19–8.90) | 6.49 (4.14–10.23) | 6.64 (4.17–8.02) |
| **Salivary cortisol nmol/L sample 2**＊ | 9.87 (7.18–13.26) | 10.61 (6.49–15.02) | 9.37 (7.42–11.94) |
| **Salivary cortisol nmol/L sample 3**＊ | 2.60 (1.80–3.95) | 2.68 (1.94–4.62) | 2.31 (1.42–3.20) |
| **Salivary cortisol nmol/L sample 4**＊ | 0.91 (0.59–1.51) | 0.97 (0.59–1.52) | 0.76 (0.59–1.50) |
| **AUCg salivary cortisol** | 46.60 (17.74) | 48.58 (18.82) | 43.96 (16.06) |

*Presented as medians (inter quartile ranges)

** $p < 0.01$.

**BMI** = body mass index; **SES** = socioeconomic status; **DHEA** = dehydroepiandrosterone; **AUCg** = day area under the curve with respect to ground cortisol levels.

## Childhood adversity and cortisol, DHEA and cortisol/DHEA ratio in hair

If adversity was analyzed as a total score, and not according to person- and environment related events, no significant associations with hair outcomes were found (all $p \geq 0.05$). However, if we split the analyses into total scores of person- and environment related events, significantly higher DHEA levels were observed in girls with more person-related childhood adversity, but not in girls with more environment-related childhood adversity (Table 4). In boys, no significant association was observed between person-related childhood adversity and DHEA levels. DHEA levels showed a trend towards being significantly higher in boys with more environment-related childhood adversity ($p = 0.07$). Hair cortisol levels were significantly higher in boys with more person-related childhood adversity (Table 4).

## Daily cortisol output in saliva

The AUCg analyses showed no differences in daily cortisol output between boys or girls with and without person-related childhood adversity. There was a trend towards a significantly lower daily cortisol output in boys with environment-related childhood adversity (b = -9.37; SE = 5.07; $p = 0.08$) compared to boys without environment-related childhood adversity (Figs 2–5).

## Discussion

Our findings demonstrated that person- and environment related childhood adversity were differentially associated with HPA-axis functioning in adolescents in a sex-specific manner.

**Table 3. Description of adverse events and frequency of experiencing such an event for girls and boys, n (%), and mean total scores (SD).**

|  | Girls (n = 116) | Boys (n = 74) |
|---|---|---|
| **Person-related events** |  |  |
| Total score | 12.9 (35.2) | 14.4 (30.6) |
| Severe illness or accident | 14 (12%) | 14 (19%) |
| Physically handicapped | 2 (2%) | 1 (1%) |
| Moved to other family | 2 (2%) | 0 |
| Victim of violence | 3 (3%) | 1 (1%) |
| Threatened with a weapon | 1 (1%) | 0 |
| **Environment-related events** |  |  |
| Total score | 79.7 (92.2) | 70.1 (80.9) |
| Severe illness or accident of family member | 24 (21%) | 18 (24%) |
| Severe illness or accident of good friend | 8 (7%) | 3 (4%) |
| Physically handicapped family member | 1 (1%) | 2 (3%) |
| Drugs addiction of family member | 4 (4%) | 1 (1%) |
| Death of a parent | 3 (3%) | 2 (3%) |
| Death of a sibling | 1 (1%) | 0 |
| Death of someone you cared about | 42 (37%) | 25 (34%) |
| Parental unwanted unemployment | 9 (8%) | 11 (15%) |
| Parental divorce or separation | 8 (7%) | 5 (7%) |
| Family member moved | 5 (4%) | 1 (1%) |
| Family member taken into custody | 1 (1%) | 1 (1%) |

Neither person- nor environment related childhood adversity was associated with an altered cortisol/DHEA ratio in hair. However, in girls, person-related childhood adversity was associated with higher hair DHEA levels, while environment-related childhood adversity was not associated with any of the HPA-axis measures. In boys, person-related childhood adversity was associated with higher hair cortisol levels. Also, for environment-related childhood adversity a trend was observed for lower daily cortisol output measured in saliva and higher DHEA levels in hair.

To the best of our knowledge, our study is the first to make a distinction in person and environment related adversity while studying associations between childhood adversity and HPA-

**Table 4. Effect sizes of regression models with person- and environment related childhood adversity as determinants and levels of DHEA, cortisol and cortisol/DHEA ratio in hair as outcome variables.**

|  | Girls |  | Boys |  |
|---|---|---|---|---|
|  | Person-related childhood adversity | Environment-related childhood adversity | Person-related childhood adversity | Environment-related childhood adversity |
| **DHEA pg/mg last month** | 0.079 (0.032)* | -0.007 (0.013) | -0.009 (0.134) | 0.093 (0.050)† |
| **Cortisol pg/mg last month** | -0.001 (0.005) | -0.001 (0.002) | 0.016 (0.007)* | -0.002 (0.003) |
| **Cortisol/DHEA last month** | -0.0002 (0.001) | -0.000052 (0.0003) | -0.002 (0.003) | -0.001 (0.001) |

Results are displayed as B (standard error)

*$p$ value $< 0.05$

†$p$ value $< 0.1$; all analyses were adjusted for age, BMI and PDS score.

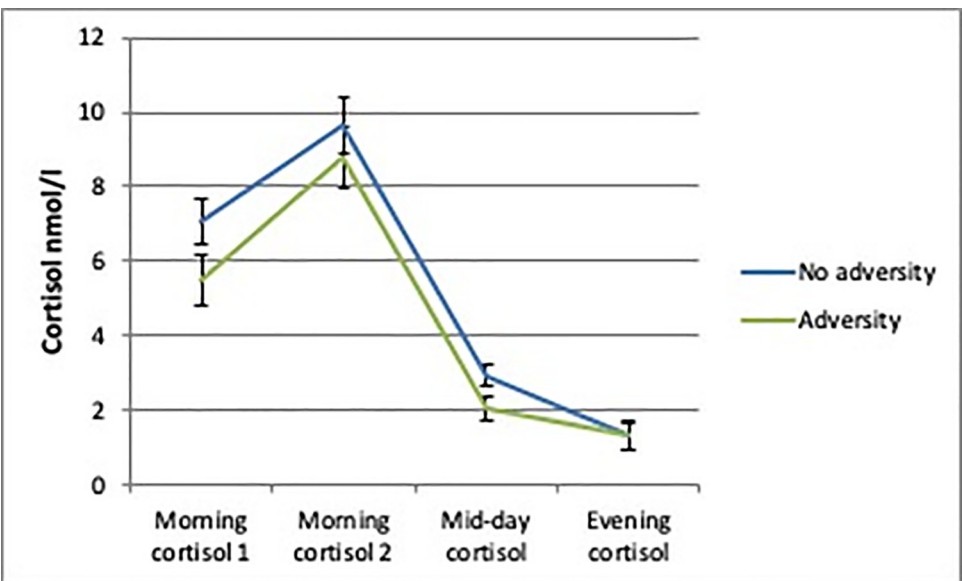

**Fig 2. Daily cortisol output from saliva for boys with and without environment-related adversity.**

axis outcomes. If the person- and environment-related events were combined in a total adverse event score, no associations were found with either cortisol or DHEA levels, which underlines the importance of studying person- and environment-related events separately. Our findings showed that person-related childhood adversity, but not environment-related adversity, was associated with altered HPA-axis functioning in girls. In boys, both person- and environment-related childhood adversity were associated with altered HPA-axis functioning. Although some previous studies suggested that different types of childhood adverse events could be associated with mental health in a sex-specific manner [5, 6, 38, 39], this is the first study showing

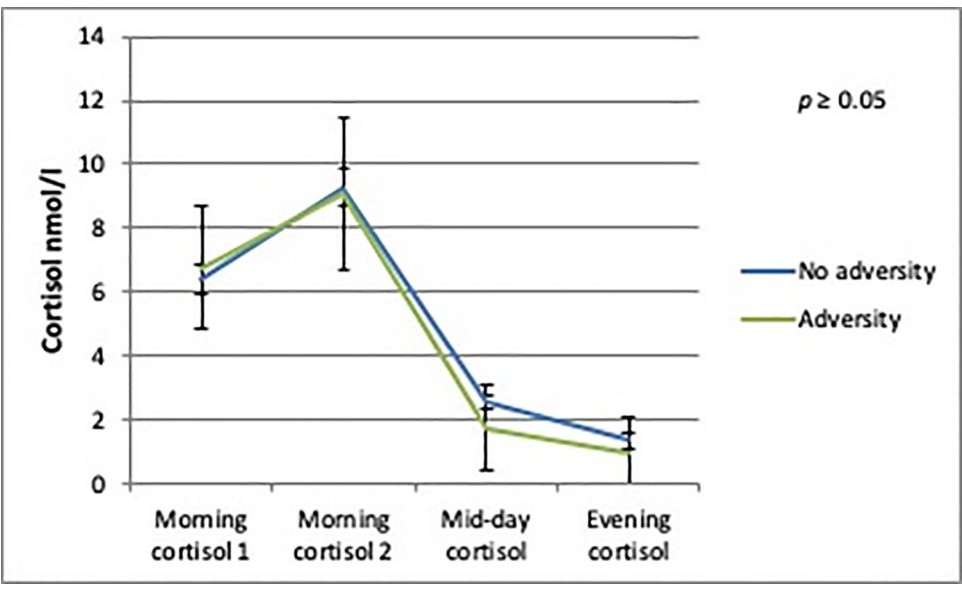

**Fig 3. Daily cortisol output from saliva for boys with and without person-related adversity.**

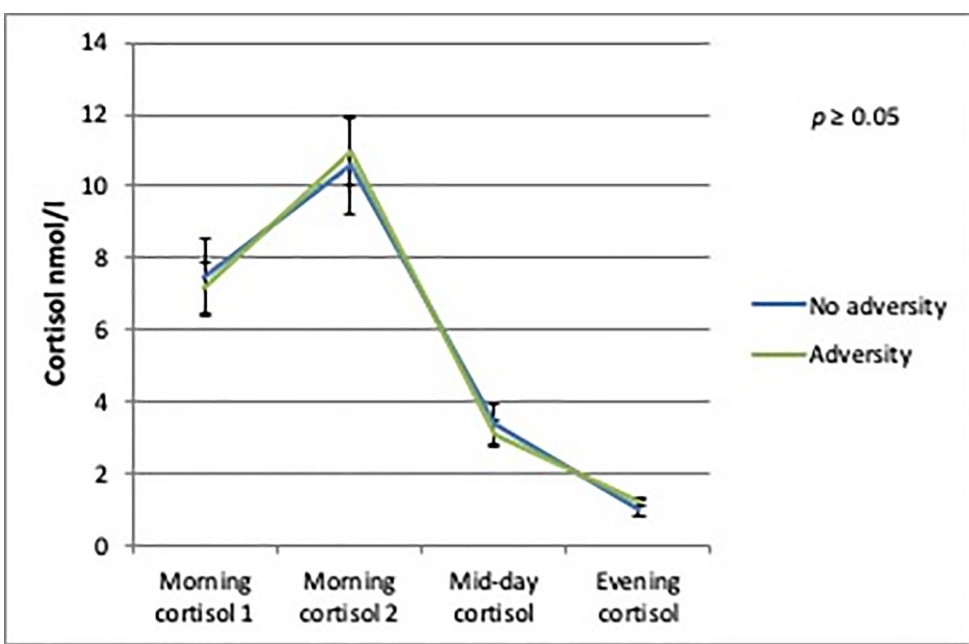

**Fig 4. Daily cortisol output from saliva for girls with and without environment-related adversity.**

that person- and environment-related childhood adversity is associated with levels of cortisol and DHEA in a sex-specific manner.

Previous research has suggested a sex-specific association between non-specific childhood adversity and altered HPA-axis functioning [40]. A potential explanation for the sex-specific associations between childhood adversity and altered HPA-axis functioning might be the differences in puberty development between girls and boys. The functioning of the HPA-axis changes during puberty from relatively low to increasing levels of both cortisol and DHEA,

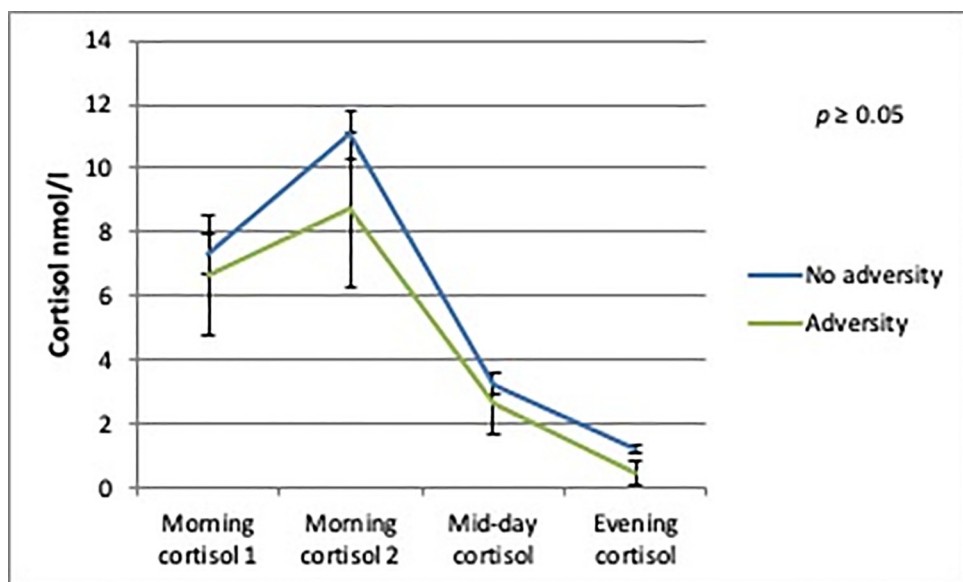

**Fig 5. Daily cortisol output from saliva for girls with and without person-related adversity.**

and these changes tend to occur earlier in girls compared to boys [41, 42]. Childhood adversity may lead to different HPA-axis functioning in girls and boys, such that the brain and endocrine system react differently to the environment during pubertal development [40]. Previous research has shown that early life adversity is associated with altered cortisol day output in boys, which is in line with our findings [43]. Our results add that the type of event is also related to these sex-specific associations. In one study, boys experienced more behavioral problems after parental divorce compared to girls, which is consistent with the idea that environment-related childhood adversity affects boys differently than girls [44]. The differences in levels of cortisol and DHEA we found are relatively small compared to effects of pubertal status but could still have important implications for further development and future mental health outcomes. These findings together suggest that the increased susceptibility for mental health and behavioral problems through altered HPA-axis functioning after childhood adverse events depend on both sex and type of event, although more research is necessary to confirm and further unravel these associations.

DHEA is a relatively new marker of mental health and recent studies have suggested a potential important role of DHEA in childhood adversity research, such that cortisol and DHEA levels could be used as predictors of mental health and behavioral problems after childhood adversity exposure [27, 40]. The present study showed that there are sex-specific differential associations between different types of childhood adversity and levels of DHEA. Based on our results it seems that girls who have experienced person-related adverse events during childhood and boys who have experienced environment-related adverse childhood events have prolonged increased levels of DHEA, although the latter was not statistically significant. Increased DHEA levels have been shown in children with anxiety and obsessive-compulsive symptoms and in adult patients with major depressive disorder and addictive disorder [20, 45, 46]. Taken together with the results from our study, childhood adversity may be linked to increased DHEA levels in girls, with potential long-term negative effects on mental wellbeing.

In a meta-analysis, childhood adversity was associated with increased cortisol levels in hair in both girls and boys, whereas our findings only showed increased cortisol levels in hair in boys who experienced adversity [47]. Our finding of a trend towards an association between environment-related childhood adversity and lower daily cortisol output in boys is in line with previous research focusing on non-specific childhood adversity and childhood daily cortisol output [24, 43]. Hair cortisol is a measure of long-term cortisol levels whereas salivary cortisol represents a short-term measure that is highly adaptive to acute changes. Previous research indicates that these two different measures may not always be in line with each other and especially salivary cortisol measured on a single day has been shown to be unassociated with hair cortisol levels [48, 49]. We found increased levels of cortisol in hair in boys exposed to person-related childhood adversity and a trend for lower daily cortisol output measured in saliva in boys exposed to environment-related childhood adversity. Because the results regarding daily cortisol output were not statistically significant they should be interpreted with caution. Collectively, these findings suggest that long-term and short-term measures of cortisol may show different patterns in relation to childhood adversity, also dependent on the type of adversity, especially when salivary cortisol is measured on a single day.

Contrary to previous research, we did not find an association between person- and environment related childhood adversity and alterations of the cortisol/DHEA ratio. This null finding may be related to the small variation in cortisol/DHEA ratio in our sample, or the greater severity of stress experienced in childhood in previous research describing effects on the cortisol/DHEA ratio [18, 40]. In addition, cortisol/DHEA ratio was derived from plasma and saliva in these studies, whereas in our study cortisol/DHEA ratio was derived from hair, which may have resulted in less variation in concentrations. Based on previous research though, the

cortisol/DHEA ratio is an important measure of neuroendocrine functioning [18], and additional research is needed to assess the sex-specific association between person- and environment related childhood adversity and this ratio.

Strengths of this study included the measurement of both cortisol and DHEA levels in hair, and daily salivary cortisol output, giving information about HPA-axis activity over longer periods of time. In line with previous studies showing a positive correlation between cortisol levels from hair and saliva [50–52], we also observed a positive correlation between the hair cortisol sample representative for the last month and the mid-day saliva sample, although measurements at other time points were not related. A limitation of this study is that we could not adjust for smoking, as reliable information was only available for ever having smoked and not for current smoking. Since there are associations between smoking and DHEA and cortisol, adjustment for current smoking would have been preferred [53]. Another limitation concerns the analysis of hair to determine cortisol and DHEA levels; in boys or girls with short hair it was not possible to take a hair sample or only a short piece of hair. Furthermore, the smaller number of adolescents participating in the saliva sample part of the study (48%) limited the statistical power for finding differences in the daily cortisol output. This particular group did not seem to be a selective group though as the responder analyses showed no differences between participants in the saliva sample part of the study and participants in the hair sample part of the study. Lastly, limiting saliva sampling to one day reduced the certainty of the cortisol day output results, since multiple sampling days may provide a more reliable estimate [54].

Despite these limitations, the study revealed differential associations between person- and environment related childhood adversity and functioning of the HPA-axis in girls and boys. The results from this study contribute to the understanding of the associations between childhood adversity and neuroendocrine functioning and emphasize that different types of childhood adversity may not only be linked to cortisol but also DHEA levels in a sex-specific manner. This may partly explain the differences between boys and girls in the subsequent development of mental health and behavioral problems. Future research is recommended to assess sex-specific associations between different types of adverse events and later mental wellbeing.

## Supporting information

**S1 File.**
(SAV)

## Author Contributions

**Conceptualization:** Lotte van Dammen, Susanne R. de Rooij, Anja C. Huizink.

**Data curation:** Susanne R. de Rooij, Pia M. Behnsen.

**Formal analysis:** Lotte van Dammen.

**Investigation:** Anja C. Huizink.

**Project administration:** Susanne R. de Rooij.

**Software:** Lotte van Dammen.

**Supervision:** Susanne R. de Rooij, Anja C. Huizink.

**Writing – original draft:** Lotte van Dammen.

**Writing – review & editing:** Lotte van Dammen, Susanne R. de Rooij, Pia M. Behnsen, Anja C. Huizink.

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
