## [Decision Letter · Decision Letter 0]

27 Feb 2020

PONE-D-19-35694

Sex-specific associations between person and environment-related childhood adverse events and levels of cortisol and DHEA in adolescence

PLOS ONE

Dear Dr. van Dammen,

Thank you for submitting your manuscript to PLOS ONE. After careful consideration, we feel that it has merit but does not fully meet PLOS ONE’s publication criteria as it currently stands. Therefore, we invite you to submit a revised version of the manuscript that addresses the points raised during the review process.

We would appreciate receiving your revised manuscript by Apr 12 2020 11:59PM. To enhance the reproducibility of your results, we recommend that if applicable you deposit your laboratory protocols in protocols.io, where a protocol can be assigned its own identifier (DOI) such that it can be cited independently in the future. For instructions see: http://journals.plos.org/plosone/s/submission-guidelines#loc-laboratory-protocols

We look forward to receiving your revised manuscript.

Kind regards,

Alexandra Kavushansky, PhD

Academic Editor

PLOS ONE

Additional Editor Comments (if provided):

This is an interesting and potentially important report. There are, however, several issues to address.

A lot of important information is missing. Such as: 1. How LC-MS/MS and ELIZA analyses were performed? 2. How the normality of the data distribution was tested? 3. Was the person performing the analyses blind to the experiment (e.g. participants` groups) details? 4. How was the participants` conformity to following the instructions for gathering and preserving the saliva samples verified? How were the samples delivered to the lab? (frozen?) 5. Why only half of the initial subjects` sample was tested for the saliva measures?

There are several language errors, such as: "…significant lower levels (line 234)", " boys compared than girls (line 270)", etc.

Graphs do not show neither significant results, nor trends.

Journal Requirements:

Reviewers' comments:

Reviewer's Responses to Questions

**Comments to the Author**

1. Is the manuscript technically sound, and do the data support the conclusions?

Reviewer #1: Yes

2. Has the statistical analysis been performed appropriately and rigorously? 

Reviewer #1: Yes

3. Have the authors made all data underlying the findings in their manuscript fully available?

Reviewer #1: Yes

4. Is the manuscript presented in an intelligible fashion and written in standard English?

Reviewer #1: Yes

5. Review Comments to the Author

Reviewer #1: This study reports differential associations between person and environment related childhood adversity and functioning of the HPA-axis in girls and boys. The main findings indicate that in girls, person-related childhood adversity was associated with higher hair DHEA levels, while environment-related childhood adversity was not associated with any of the HPA-axis measures. In boys, person-related childhood adversity was associated with higher hair cortisol levels. The introduction gives and adequate background and clearly describes the state of art in the field. The methods are sound and relevant to the topic being researched. Results are mostly clear and well written, and the discussion is properly presented. However, some minor modifications are suggested, as follows:

1.Results Section

1.1 It would be helpful for the reader if the significant differences of table 2 were clearly indicated, along with the respective p value.

1.2 Authors use terms like similar and comparable, instead of no significant differences. Please indicate the p values to clearly indicate that comparable means not significantly different.

1.3 The term “trend” appears in the results and discussion section, but the p value is not shown in the tables or text. Please insert the p value.

2 Discussion section

2.1 Please explain the sentence below in more detail in the discussion section.

“These findings collectively indicate that different types of childhood adversity may be linked to blunted daily cortisol output, but overall prolonged elevated levels of cortisol, suggesting HPA-axis dysregulation, with possible future associations with impaired mental wellbeing.”

How can a daily blunted cortisol lead to an increase in hair cortisol? Would it not be expected that lower daily cortisol levels are associated with lower hair cortisol? How can the results of previous research focusing on non-specific childhood adversity and childhood daily cortisol output be reconciled with the higher hair cortisol shown in this study?

2.2 Are the size effects found in this study for the association of DHEA with person-related childhood adversity in girls, and for the association of hair cortisol and person-related childhood adversity in boys, of clinical relevance? Please compare with the size effect of the association of DHEA and cortisol with the pubertal stage and with other studies, if possible.

2.3 Was the association of the hair cortisol with the AUCg evaluated? If not, please explain why. If it was evaluated, please state the results clearly in the text.

6. PLOS authors have the option to publish the peer review history of their article (what does this mean?). If published, this will include your full peer review and any attached files.

Reviewer #1: Yes: Elke Bromberg

---

## [Author Response · Author response to Decision Letter 0]

27 Mar 2020

Dear Dr. Alexandra Kavushansky,

We would like to thank the editor and reviewer for their time and the useful comments regarding the manuscript ‘Sex-specific associations between person and environment-related childhood adverse events and levels of cortisol and DHEA in adolescence’. Please find our answers to the comments and questions below.

Editor comments:

1. How LC-MS/MS and ELIZA analyses were performed? 

We have clarified the methods in the methods section. The LC-MS/MS assays were performed based on the method as described in the following paper: Gao, W., Stalder, T., Foley, P., Rauh, M., Deng, H., & Kirschbaum, C. (2013). Quantitative analysis of steroid hormones in human hair using a column-switching LC–APCI–MS/MS assay. Journal of Chromatography B., 928, 1–8. doi:10.1016/j.jchromb.2013.03.008

ELIZA: Analysis of cortisol concentrations was carried out by means of salivary cortisol immunoassay with time-resolved fluorescence endpoint detection. (as described in Dressendorfer et al., 1992)

All analyses were performed by Dresden Labservices in Germany.

2. How the normality of the data distribution was tested? 

Normality was assessed by visual inspection of histograms and Kolmogorov-Smirnov tests. We have added this information in the statistical analyses section.

3. Was the person performing the analyses blind to the experiment (e.g. participants` groups) details? 

No, the person performing the analyses was not blind to the groups since the groups were not defined before the study started but based on outcomes of the study and by the same person who performed the analyses. Groups were defined before analysis of the lab outcomes though, so we are sure that this has not affected the study outcomes.

4. How was the participants` conformity to following the instructions for gathering and preserving the saliva samples verified? How were the samples delivered to the lab? (frozen?) 

Participants received detailed instructions, in both text and pictures, about the collection of the saliva samples. They were asked to provide the actual time of collecting the sample, also if the time deviated from the time they were supposed to collect the sample. However, we cannot be sure if they truly did so. Conformity to the instructions was not specifically verified.

Samples were sent unfrozen by mail and were immediately stored in the freezer until the assays were performed. We have added this information to the methods section.

5. Why only half of the initial subjects` sample was tested for the saliva measures?

Only half of the initial group was invited to participate in the second part of the study; the saliva sample part. There was not enough funding available to assay saliva samples for the entire group.

6. There are several language errors, such as: "…significant lower levels (line 234)", " boys compared than girls (line 270)", etc.

We thank the editor for spotting these mistakes and changed it to “significantly lower levels” and “boys compared to girls”. We have also thoroughly read the manuscript again looking for other language errors but think there are no other errors.

7. Graphs do not show neither significant results, nor trends.

We have included the P values in the graphs.

We have attached the figures separately.

We have included a dataset.

We have added an ORCID ID for the corresponding author.

We have removed the phrase “data not shown” and replaced it with the results.

Reviewer comments:

1.Results Section

1.1 It would be helpful for the reader if the significant differences of table 2 were clearly indicated, along with the respective p value.

We have marked the significant differences and included the P value.

1.2 Authors use terms like similar and comparable, instead of no significant differences. Please indicate the p values to clearly indicate that comparable means not significantly different.

We have changed the terminology to clearly indicate the level of significance. 

1.3 The term “trend” appears in the results and discussion section, but the p value is not shown in the tables or text. Please insert the p value.

P value has been inserted.

2 Discussion section

2.1 Please explain the sentence below in more detail in the discussion section.

“These findings collectively indicate that different types of childhood adversity may be linked to blunted daily cortisol output, but overall prolonged elevated levels of cortisol, suggesting HPA-axis dysregulation, with possible future associations with impaired mental wellbeing.”

How can a daily blunted cortisol lead to an increase in hair cortisol? Would it not be expected that lower daily cortisol levels are associated with lower hair cortisol? How can the results of previous research focusing on non-specific childhood adversity and childhood daily cortisol output be reconciled with the higher hair cortisol shown in this study?

Previous research has shown that childhood adversity is linked to blunted diurnal cortisol levels but also to increased cortisol levels in response to a stressor. This could be an explanation for our findings of blunted daily cortisol levels and overall elevated levels of cortisol (measured in hair).

2.2 Are the size effects found in this study for the association of DHEA with person-related childhood adversity in girls, and for the association of hair cortisol and person-related childhood adversity in boys, of clinical relevance? Please compare with the size effect of the association of DHEA and cortisol with the pubertal stage and with other studies, if possible.

We have compared the effect size of the association between childhood adversity and DHEA and cortisol to the effect size of pubertal stage and described the results in the discussion section.

2.3 Was the association of the hair cortisol with the AUCg evaluated? If not, please explain why. If it was evaluated, please state the results clearly in the text.

Yes, it was evaluated, and the results are described in the descriptive analyses section in the results section (lines 217-220).

---

## [Decision Letter · Decision Letter 1]

10 Apr 2020

PONE-D-19-35694R1

Sex-specific associations between person and environment-related childhood adverse events and levels of cortisol and DHEA in adolescence

PLOS ONE

Dear Dr. van Dammen,

Thank you for submitting your manuscript to PLOS ONE. After careful consideration, we feel that it has merit but does not fully meet PLOS ONE’s publication criteria as it currently stands. Therefore, we invite you to submit a revised version of the manuscript that addresses the points raised during the review process.

We would appreciate receiving your revised manuscript by May 25 2020 11:59PM. To enhance the reproducibility of your results, we recommend that if applicable you deposit your laboratory protocols in protocols.io, where a protocol can be assigned its own identifier (DOI) such that it can be cited independently in the future. For instructions see: http://journals.plos.org/plosone/s/submission-guidelines#loc-laboratory-protocols

We look forward to receiving your revised manuscript.

Kind regards,

Alexandra Kavushansky, PhD

Academic Editor

PLOS ONE

Reviewers' comments:

Reviewer's Responses to Questions

**Comments to the Author**

1. If the authors have adequately addressed your comments raised in a previous round of review and you feel that this manuscript is now acceptable for publication, you may indicate that here to bypass the “Comments to the Author” section, enter your conflict of interest statement in the “Confidential to Editor” section, and submit your "Accept" recommendation.

Reviewer #1: All comments have been addressed

2. Is the manuscript technically sound, and do the data support the conclusions?

Reviewer #1: Yes

3. Has the statistical analysis been performed appropriately and rigorously? 

Reviewer #1: Yes

4. Have the authors made all data underlying the findings in their manuscript fully available?

Reviewer #1: Yes

5. Is the manuscript presented in an intelligible fashion and written in standard English?

Reviewer #1: Yes

6. Review Comments to the Author

Reviewer #1: Although all the comments were addressed, some important questions were only partially answered. Please see below:

2 Discussion section

2.1 Please explain the sentence below in more detail in the discussion section.

“These findings collectively indicate that different types of childhood adversity may be linked to blunted daily cortisol output, but overall prolonged elevated levels of cortisol, suggesting HPA-axis dysregulation, with possible future associations with impaired mental wellbeing.”

How can a daily blunted cortisol lead to an increase in hair cortisol? Would it not be expected that lower daily cortisol levels are associated with lower hair cortisol? How can the results of previous research focusing on non-specific childhood adversity and childhood daily cortisol output be reconciled with the higher hair cortisol shown in this study?

The authors did not answer the questions bellow:

- How can a daily blunted cortisol lead to an increase in hair cortisol?

- Would it not be expected that lower daily cortisol levels are associated with lower hair cortisol?

- How can the results of previous research focusing on non-specific childhood adversity and childhood daily cortisol output be reconciled with the higher hair cortisol shown in this study? Note that the question is about low daily cortisol and higher hair cortisol.

Even if there is no conclusive explanation to these questions, the authors should discuss the apparent contradictory results in daily cortisol and hair cortisol. Are there other studies, with children or other developmental stages, that show increased hair cortisol and blunted daily cortisol? These questions are important because hair and daily cortisol represent different time windows, i.e., chronic versus acute cortisol levels. Moreover, the decrease in cortisol seen in this study didn’t reach statistical significance (p=0.08). Is the conclusion about blunted cortisol levels in this study perhaps overestimated?

2.3 Was the association of the hair cortisol with the AUCg evaluated? If not, please explain why. If it was evaluated, please state the results clearly in the text.

Below is the answer given by the authors:

“Yes, it was evaluated, and the results are described in the descriptive analyses section in the results section (lines 217-220).”

Please note that the question is about the association of the hair cortisol with the AUCg, whereas the answer refers to the comparison of hair cortisol samples and the separate saliva cortisol samples (not to the AUCg, as asked). The analysis of the association between AUCg and hair cortisol would perhaps be helpful to the answer of comment 2.1.

7. PLOS authors have the option to publish the peer review history of their article (what does this mean?). If published, this will include your full peer review and any attached files.

Reviewer #1: Yes: Elke Bromberg

---

## [Author Response · Author response to Decision Letter 1]

23 Apr 2020

Dear Dr. Alexandra Kavushansky,

We would like to thank the reviewer for their time and the useful comments regarding the manuscript ‘Sex-specific associations between person and environment-related childhood adverse events and levels of cortisol and DHEA in adolescence’. Please find our answers to the comments and questions below.

Reviewer comments:

2 Discussion section

2.1 Please explain the sentence below in more detail in the discussion section.

“These findings collectively indicate that different types of childhood adversity may be linked to blunted daily cortisol output, but overall prolonged elevated levels of cortisol, suggesting HPA-axis dysregulation, with possible future associations with impaired mental wellbeing.”

We agree that this sentence needs more explanation and we have changed the discussion section regarding this subject.

How can a daily blunted cortisol lead to an increase in hair cortisol? Would it not be expected that lower daily cortisol levels are associated with lower hair cortisol? 

We agree that it would be expected that lower daily cortisol levels are associated with lower hair cortisol levels. We found additional literature regarding the correspondence between hair and salivary cortisol levels and in these two papers no significant correlation was observed between hair cortisol levels and salivary cortisol measured on a single day (including the AUCg). Based on these papers we changed the discussion section regarding this point.

How can the results of previous research focusing on non-specific childhood adversity and childhood daily cortisol output be reconciled with the higher hair cortisol shown in this study?

See the discussion section lines 304-315 for our changes.

The authors did not answer the questions bellow:

- How can a daily blunted cortisol lead to an increase in hair cortisol?

There is a weak link between hair cortisol and cortisol measured in saliva on a single day. In order to answer this question, we need to include salivary cortisol levels measured on multiple days, but these samples were not collected, unfortunately. We have also discussed this as a limitation to our study (see Strengths and Limitations paragraph).

- Would it not be expected that lower daily cortisol levels are associated with lower hair cortisol?

Yes, see above.

- How can the results of previous research focusing on non-specific childhood adversity and childhood daily cortisol output be reconciled with the higher hair cortisol shown in this study? 

See above.

Note that the question is about low daily cortisol and higher hair cortisol.

Even if there is no conclusive explanation to these questions, the authors should discuss the apparent contradictory results in daily cortisol and hair cortisol. Are there other studies, with children or other developmental stages, that show increased hair cortisol and blunted daily cortisol? These questions are important because hair and daily cortisol represent different time windows, i.e., chronic versus acute cortisol levels. Moreover, the decrease in cortisol seen in this study didn’t reach statistical significance (p=0.08). Is the conclusion about blunted cortisol levels in this study perhaps overestimated?

We agree that the trend level significance should be interpreted with more caution. See the discussion section for our changes in lines 310-312.

2.3 Was the association of the hair cortisol with the AUCg evaluated? If not, please explain why. If it was evaluated, please state the results clearly in the text.

Below is the answer given by the authors:

“Yes, it was evaluated, and the results are described in the descriptive analyses section in the results section (lines 217-220).”

Please note that the question is about the association of the hair cortisol with the AUCg, whereas the answer refers to the comparison of hair cortisol samples and the separate saliva cortisol samples (not to the AUCg, as asked). The analysis of the association between AUCg and hair cortisol would perhaps be helpful to the answer of comment 2.1.

See lines 219-220 in the results section for the AUCg correlation with hair cortisol: “hair cortisol over the last month was not associated with AUCg cortisol levels (r = 0.11, p = 0.28).”

---

## [Editor Report · Decision Letter 2]

12 May 2020

Sex-specific associations between person and environment-related childhood adverse events and levels of cortisol and DHEA in adolescence

PONE-D-19-35694R2

Dear Dr. van Dammen,

We are pleased to inform you that your manuscript has been judged scientifically suitable for publication and will be formally accepted for publication once it complies with all outstanding technical requirements.

With kind regards,

Alexandra Kavushansky, PhD

Academic Editor

PLOS ONE
---

## [Editor Report · Acceptance letter]

20 May 2020

PONE-D-19-35694R2 

Sex-specific associations between person and environment-related childhood adverse events and levels of cortisol and DHEA in adolescence 

Dear Dr. van Dammen:

I am pleased to inform you that your manuscript has been deemed suitable for publication in PLOS ONE. Congratulations! Your manuscript is now with our production department. 

With kind regards,

on behalf of

Dr. Alexandra Kavushansky 

Academic Editor

PLOS ONE